# Exploring for-profit healthcare providers' perceptions of inclusion in the Zambia National Health Insurance Scheme: A qualitative content analysis

Kwangaika Mwala Sinjela[1], Warren Mukelabai Warren Simangolwa[2], Lindsey Hehman[2], Mpuma Kamanga[3], Wesley Kapaya Mwambazi[4], Jesper Sundewall[5,6]*

1 Faculty of Medicine, Lund University, Malmö, Sweden, 2 Clinton Health Access Initiative, Lusaka, Zambia, 3 National Health Insurance Management Authority (NHIMA), Lusaka, Zambia, 4 Ministry of Health, Lusaka, Zambia, 5 Division of Social Medicine and Global Health, Lund University, Malmö, Sweden, 6 University of KwaZulu-Natal, Durban, South Africa

☉ These authors contributed equally to this work.
* jesper.sundewall@med.lu.se

**Data Availability Statement:** All relevant data are within the manuscript and its Supporting Information files.

## Abstract

### Background

In 2019, Zambia introduced the national health insurance (NHI) as a healthcare financing strategy to increase universal access to health care services. The private health sector can complement public sector providers as service providers under the NHI. As such, the NHI Management Authority seeks to accredit for-profit private healthcare facilities in the NHI. Ascertaining factors that influence private-for-profit health providers to participate in the NHI is essential, but the evidence is lacking. In this study, we aimed to explore and characterize perceptions and experiences of for-profit private hospitals, dental clinics, eye clinics, diagnostic centres, and pharmacies regarding their inclusion in the NHI.

### Methods

We conducted in-depth interviews with owners or management officers of purposively sampled private health care providers in Lusaka, Zambia (n = 22) between May and June 2020. Qualitative content analysis was used to analyse data.

### Results

The findings highlight low awareness of the NHI among providers and a need to understand the NHI. Providers revealed their positions and views on the accreditation process and payment arrangements and stated that their participation would complement the NHI. They also cited conditions to participate in the NHI, highlighted opportunities and challenges of engaging in the NHI, and expressed a need for sustainable ways of governing the scheme.

**Funding:** KMS was funded to conduct the study by the Clinton Health Access Initiative (CHAI, https://clintonhealthaccess.org). The funders had no role in study design, data collection and analysis, decision to publish, or preparation of the manuscript.

**Competing interests:** The authors have declared that no competing interests exists.

## Conclusion

The assessment of health providers' inclusion in the NHI scheme is multifaceted. The results of this study surfaced factors such as raising awareness on the NHI among providers and how their concerns on aspects such as payments can be considered as inputs to enlighten consensual agreements between the NHI authority and health providers in leveraging the private health sector. Private providers' concerns must be further understood and considered as the NHI strives to include this group as health care providers in the scheme.

## Introduction

On the road towards universal health coverage (UHC), countries continue to scale-up or establish nation-wide insurance systems with increasing attention to the role of private health care providers [1,2]. Private for-profit healthcare providers (private providers) can be a significant resource in health systems designed to achieve UHC [2]. Studies have shown that private providers complement the public with unique services, increased responsiveness of the insurance system and distribution of healthcare services to the public [1–6]. Engaging the private providers in the national health insurance may also enhance equitable access to private healthcare and improve equality in service provision [1–3,7]. As such, public-private partnerships can benefit the national health insurance and contribute to achieving UHC objectives [8].

In efforts to reach UHC by 2030, Zambia implemented a National Health Insurance (NHI) scheme (NHIS) a healthcare financing strategy. The NHIS was asserted to as law in 2018, operationalized in 2019, and became accessible by registered NHI members effective 1st February 2020. The NHI is operated and managed by the National Health Insurance Management Authority (NHIMA) and it is financed by a 1% NHI statutory premiums, payroll-based from employees and employers in the formal sector, and 1% of the declared income from those who are self-employed [9–11]. Those over 65 and under 18 years old are exempted from pay NHI contributions.

NHIMA accredited 129 level 1–3 government, faith-based/mission and non-governmental organization (NGO) owned health facilities by December 2019 and aims to accredit private providers in its second phase of NHI implementation [10]. Target private providers are pharmacies, diagnostic facilities (laboratory and imaging centres), dental clinics, eye clinics, and hospitals. The scope of including these facilities into the scheme is guided by the facility's geographical distribution (Fig 1), infrastructure, catchment population, services and products including quality, efficiency, bulk and availability, and how best these facilities would complement the NHI's health benefits package (HBP) to cover the limitations in the public sector [10].

It is vital to determine factors that influence private providers' participation in the scheme before deliberations for their integration [1]. Studies done on NHI systems of Sub-Saharan African countries such as Nigeria, Ghana, Rwanda, Kenya, and Uganda have explored certain factors that influence the inclusion of private providers into the NHI scheme [1,3–5,7,12–17]. These studies have shown various NHI contractual arrangements, role of the private providers in NHI, and opportunities and challenges associated with their inclusion in NHI systems [1,3–5,7,12–19]. However, all these studies were conducted in already existing insurance systems [1,15,20]. Also, there is a paucity of evidence in Zambia on the for-profit health market to inform evidence-based decision-making in the NHI [6,21]. To the best of our knowledge, no

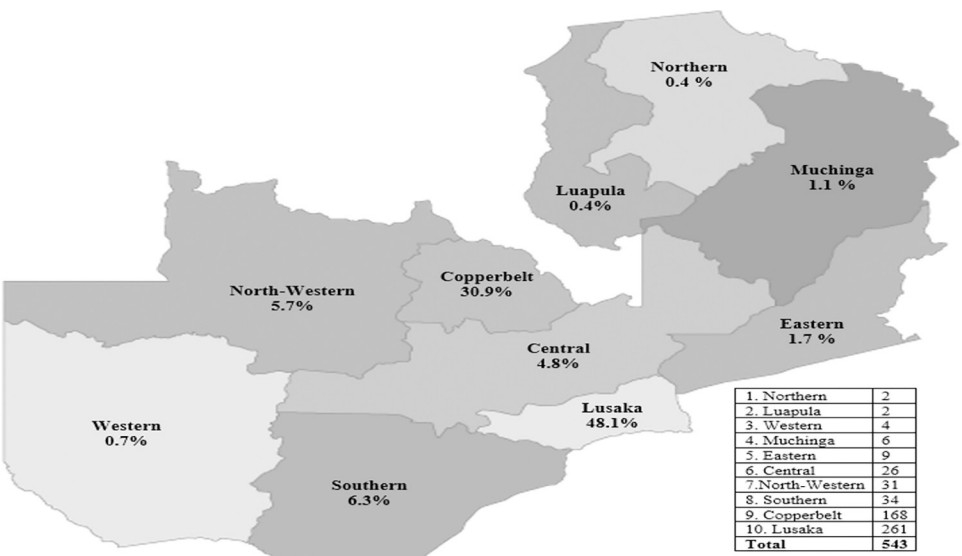

**Fig 1. Distribution of accredited private health facilities segregated by provinces in Zambia for the year 2019 [24].**

study has focused on private providers' perspectives before their involvement in the insurance scheme, and this is further exacerbated by limited global a priori studies focused on private providers' perspectives to their inclusion in the NHI [2]. As such, investigating factors that influence private providers would be necessary to enable efficient resource allocation in the NHIS.

Our study aimed to explore and characterize experiences and perceptions of for-profit private hospitals, dental clinics, eye clinics, diagnostic centres and pharmacies on their inclusion in the NHI scheme in Zambia. More specifically, we sought to better understand their awareness about the scheme, what value private providers would add to the NHI, their views regarding applying for accreditation, payment arrangements and management and security of information in the NHI, and the benefits and challenges they anticipate, based on experience with private insurance companies (PICs) and regulatory agencies.

## Methods

### Study design and setting

Interviews were conducted in Lusaka, Zambia's most densely populated and urban province [22,23]. Lusaka province was chosen as the study area because of having the highest proportion of private providers in the country. The city has an estimated 48% of all hospitals, clinics and diagnostic centres and about 66% of retail licensed pharmacies [24,25].

### Selection of facilities and participants

The sample consisted of 22 participants from private provider facilities determined by stratified purposeful sample selection methods [26–29]. Interviews were conducted among class A hospitals, Class C clinics and Class D diagnostic centres shortlisted from the Health Professions Council of Zambia (HPCZ) list of active and accredited private health care providers [24], and pharmacies from the Zambia Medicines Regulatory Authority (ZAMRA) list of registered retail and hospital community pharmacies and chemists [25]. Five informants from 10 selected pharmacies were interviewed, 4 from 7 selected diagnostic centres, 4 from eye clinics and 4 from dental clinics from 17 selected clinics and 5 informants were interviewed from a

**Fig 2. Process for stratified purposeful sample selection.**

selection of 22 tertiary hospitals in Lusaka (Fig 2). All hospitals and clinics had partnerships with private insurance companies (PICs).

Four informants owned the facilities and 16 were either administrators, directors, managers or human resource (HR) officers of the facilities, with the other two being consultants. Fifteen facilities had been operating for more than 7 years.

## Data collection

In-depth interviews were conducted between 15 May-8 June 2020. The interview sessions included 19 face-to-face and 3 online meetings, lasted between 20–40 min, were recorded and involved taking notes relevant for analysis. The interview guide used (S1 Appendix) was developed based on key issues highlighted in the literature and explored experiences, perceptions and attitudes toward the NHIS, its implementation and NHIMA's plans to involve private providers in the NHI [1,6,21]. It was also reviewed by study supervisors for content and construct validity and was piloted to test the adequacy, relevancy and to ensure that data generated reflected the research aim.

## Data analysis

Recorded interviews were transcribed verbatim, corrected for errors, shortened to meaning units and bracketed into condensed meaning units. Codes were then created from the 22 transcripts, merged in each facility type and abstracted into code groups, sub-categories and subsequently into categories. Our inductive analytical approach combined the qualitative content analysis steps suggested by Dahlgren [29] and Graneheim and Lundman [30] and was performed with the aid of MS Word and Excel [26,28–32].

## Ethical considerations

The study protocol was approved by ERES Converge Internal Review Board (IRB No. 00005948), CHAI's Scientific and Ethical Review Committee (SERC) and the Zambia National

Health Research Authority (NHRA) and it was authorized by the Ministry of Health (MoH) and Lusaka District Health Office (DHO), Zambia. Informed consent to take part in the study was received from private provider facilities' management and from all participants [33].

## Results

Private providers expressed a need to understand the NHI, revealed their positions on the accreditation process, viewed the NHI as an opportunity window and believed that their participation would complement public facilities in the NHI. Informants also expressed views on payment arrangements, conditions to participate in the NHI and a need for sustainable ways of governing the scheme.

### Understanding the NHI

Informants generally expressed limited understanding regarding NHIMA strategies to engage their facilities and expressed uncertainty about the NHI. Five informants stated that they had not known about the scheme and seventeen informants, who knew the scheme NHI through channels such as the NHI website, media, paying NHI levy for staff and receiving NHI insured clients, described the interviews in this study as their first opportunity to get more detailed information on some aspects of the NHI and the rest had not known about the scheme. All private providers also expressed eagerness for more information about the NHI.

- *"I didn't know it existed. You are the first to let me know about this national health insurance scheme." (I-20, Dental Clinic). "Of course we are getting information here and there, but we haven't been called in to be introduced to the scheme I am 100% sure that they will definitely come through even to us and we are more than ready to participate." (I-01, Dental Clinic).*

### Accrediting in the NHI

Regarding participating in the NHI scheme, one provider had applied for accreditation, eight were planning to apply, five were undecided and eight did not intend to apply. Private providers that had decided to apply for accreditation included those that had attended meetings on the NHI and those contacted on the pricing of services, medicines and equipment by NHIMA. The same providers anticipated benefits and opportunities to insured clients, to communities, as well as to their facilities. Financial security, increased choice for medical services, quick access to services, drug availability and easy access to health specialists were the main benefits seen for insured clients. They also viewed the NHI scheme to bring centralization, create financial equity, and that their participation would improve the healthcare sector, create employment, bring empowerment, and boost the economy for the country. These informants also saw their facilities to benefit because accrediting in the NHI would increase clientele, build their repute, grow business and create a stable income. It was also cited that in endeavouring to meet accreditation requirements, facilities would improve their quality of services and infrastructure.

- *"Yes, I think that [applying for accreditation] is our top priority at the moment" (I-001, Diagnostic Centre). "[There is] increase in access to specialized services and to specialists. . .. [There is] financial security to patients because they don't have to pay cash." (I-05, Hospital). "So if there is a central part [NHI scheme] somewhere and we become part of that central part, we should be able to complement the efforts of others because at the moment services are fragmented." (I-06, Eye Clinic). "There is a larger number of clientele. Our reputation will be built." (I-03, Hospital).*

Private providers that were undecided and those that did not intend to apply for accreditation sought to understand the NHI with incentives involved and revealed conditions for engaging in the NHI such as consensual pricing and payments. They linked current concerns, such as lack of trust in NHI feasibility, to limited understanding of the scheme and emphasized challenges in participating in the scheme. Pharmacies that did not intend to accredit specified to receive cash on demand (COD) or upfront payments to participate in the NHI scheme. Clinics and hospitals that did not intend to accredit reported that authorization for procedures from PICs is sometimes delayed, selective and restrictive and that there is delay or failure of reimbursements, inflexible pricing and rejections of claims, and anticipated these factors to exist in the NHI scheme as well. All private providers that did not intend to accredit also expressed fear regarding exploitation, cited inadequate revenue, showed scepticism on reaching UHC ideals through NHI, and necessitated a different set of accreditation standards from the public facilities. The same private providers anticipated security and management setbacks regarding prioritization and secure verification of client's identity cards (IDs) and profiles, as experienced with PICs. Lengthy accreditation, high annual accreditation fees, irrelevant regulations, losing records, and poor communication in the NHI were other sets of challenges anticipated.

- *"I see difficulties in people understanding the system." (I-07, Hospital). "We need more information and also we need to know of the rules involved" (I-20-, Dental Clinic). "The challenge is gonna be getting paid. . ." (I-002, Diagnostic Centre). "Pay on time. Pay before we order because of exchange rates. . .. We fear corruption., overcharging and abuse of services" (I-16, Chemist). "There are certain procedures that they chose not to cover. Secondly, the [pre-authorization] takes too long. . .. Same things with NHIMA" (I-03, Hospital). ". . .it [security] won't be enough even with the biometric. . .And it would even be worse with NHIMA because it's targeting the mass." (I-11, Dental Clinic).*

## Perceived value to the NHI

Informants from hospitals, clinics and diagnostic centres considered themselves to have a better standard of services, advanced treatment and equipment, timely and efficient services, and varied treatment options which they believed to bring complementary value the NHI. The same informants also stated to complement the public geographically, with hospitals having nation-wide and international reach. The drug store retailers reported selling certain drugs and medicines not available in public hospitals.

- *"Definitely, I would see myself as a necessity, complement. . . So a patient will be given that opportunity to choose what service, (I-01, Dental Clinic). ". . .. we offer very good service in terms of catchment area, we have patients from all over the country" (I-03, Hospital). "I complement. My drugs are different." (I-15, Chemist).*

## Views on payment arrangements and governance

The biggest concern revealed by all private providers regarded payments including the accreditation fee, reimbursement period, payment conditions and modes of payments. Informants expressed that their participation in the NHI scheme is a partnership engagement with NHIMA as with PICs and as such viewed the accreditation fees [11] as expensive and unnecessary. Hospitals, clinics and diagnostic centres stated to be comfortable with claim reimbursements or advance payment arrangements in the NHI as done in insurance with PICs. On the other hand, drug stores reported receiving COD or engaging in partnerships only with PICs

that pay in advance and wished for similar payment arrangements in the NHI. It was also anticipated that reimbursements in NHI would take a long time, even longer than the on average 30 days it takes to get reimbursed by PICs. Other key concerns informants raised on payments regarded reaching consensual payment agreements and the need to know details of the provider payment mechanisms regarding accredited public facilities.

- *"Why pay [accreditation fee]? I am not happy with the payment. Payment by a health provider partner with NHIMA is surprising." (I-19, Dental Clinic). "Payments are made after claims are submitted. Some pay upfront as [a] deposit." (I7-, Hospital). "I have given all of them [PICs] an option that I want the money upfront" (I-02, Chemist). "It takes 30 days for our claims to be paid, though some don't pay on time. This [NHIMA's reimbursement period] is too long. It should be reduced to at least a month." (I-07, Hospital).*

All providers agreed on secure and reliable verification of the client's identification and information by electronic identification such as fingerprinting, with eleven informants seeing the identification card provided for NHIS members as having adequate security features [9,11]. Hospitals, clinics and diagnostic centres reported card smart systems employed by PICs, which allow for access to clients' profile data via fingerprint scans, or calling to consult with the PIC before attending to an insured health seeker as trending verification measures and hoped to have the same system in the NHI.

- *"It [NHIS member card] is secure, but fingerprints should be added. It minimizes fraud challenges and it is secure." (I-07, Hospital).*

Concerning information processing, pharmacies expressed preferences for non-electronic processing (paper) to enter, keep and track records, insisting that paper saves time in their busy retail shops and allows for physical signing, which they viewed as a key verification feature. On the contrary, hospitals, clinics and diagnostic centres stated to be well established digitally, cited paper processing as tediousness and lacking scalability, and expressed preferences for electronic/digital management systems. They cited efficiency, popularity, transparency and traceability as advantages for electronic systems. However, they recognized some challenges associated with electronic processing such as slow internet connectivity, back-up failure, high maintenance costs and security issues, and expressed the need for reliable non-electronic verification and back-up systems.

- *"We don't have much time to use computer here. Paper is better." (I-18, Chemist). "Manual [paper] should only come in as backup in terms of downtime. But I'd prefer that everything is done electronically. One, because I think it offers more security. Two, it's also faster. Three, I think it also has higher traceability than the manual process. The manual [paper] process, it's quite laborious. . ." (I-10, Diagnostic Centre).*

## Discussion

The goal of establishing the NHI is to provide reliable and sustainable financing to the health system and achieve UHC in Zambia [34]. In this effort, it is necessary to appropriately consider priority factors such as understanding of the NHI among stakeholders, accreditation processes, financial-related arrangements and information security and management [1,3–5,12,14,15,18]. Knowing where private providers stand concerning these issues may guide inputs on general consensual agreements between the NHI purchaser and private providers to benefit both parties as well as the insured users [35].

Our findings show that private providers need to understand government initiatives that require their engagement. Providers that were undecided or did not want to participate

emphasized the need to understand the scheme. On the other hand, those found to be suffi-ciently aware and decided whether or not to accredit were those in direct contact with NHIMA. Lack of information on the NHI can be identified as a key barrier to the expansion of the scheme, thus indicating the significance of reaching providers through effective communi-cation channels. Similar to this study, findings in Kenya showed that unclear communication from the national hospital insurance fund (NHIF) resulted in providers not knowing how to address concerns and also cited lack of sensitization as proportional to little or no understand-ing on how the NHIF worked regarding services covered and the requirements for accredita-tion [1]. Establishing engagement channels with providers, such as continual dialogue through focus group discussions, interviews and other meetings, could be ways of raising awareness as opposed to leaving them to access information on their own. This could also contribute to cre-ating trust and confidence in the NHI scheme.

Our findings also highlight how private providers view their role to public providers. It has been shown in certain LMICs that private providers either offer choice to insured users and complements or supplements the public facilities [1,3,4,36]. Private clinics and hospitals have consistently been reported to offer care under shorter periods compared to the public sector [1,3,18], minimize hindrances and limitations experienced by patients served in the public sec-tor [3,37,38] and generally rank higher in patients' assessment of hospitality and courtesy of staff, cleanliness of facilities, diagnostic explanations and availability of particular medical inputs [3,39]. These aspects are associated with better quality in the private health sector and were also reported in the current study. Accrediting private providers in the NHI has also been associated with an increase in geographical and demographic coverage, and a potential to expand and reach poor communities where there is limited public delivery [36,37].

Providers reported challenges such as facing rejections delays or unpredictability of claim reimbursements and restrictions on services and products covered in their partnerships with PIC. These challenges are consistent with studies found in both LMIC and HIC settings and mostly regarded payments [1,3,4,7,20,21,36,40,41]. These issues, consistently cited to cause high levels of dissatisfaction and linked with medical service barriers and providers' "moral hazards" [15], were also obstacles reported as inevitable in the Zambia NHI. An imbalance between revenue generation and number of registered NHI users is also a noteworthy chal-lenge to mention which has also been identified where NHI systems in Africa have been exten-sively studied [1,15,42,43].

Our findings also highlight the desire for consensual agreements among private providers in their inclusion in the NHI. Reaching these agreements may mean that both purchaser and health providers compromise on certain aspects. For instance, the NHI authority may set accreditation fees in line with what private providers are willing to pay and implement smart verifications systems that may attract private providers [1]. Tailoring information processing and security systems to providers and having centralized and transparent systems may also curb doubts and motivate confidence in the NHI management authorities, as identified across NHI systems such as in Ghana, Kenya and Nigeria [1,7,14,17,18,38,42,43].

## Methodological considerations

The sampling process began with stratified selection based on the facility types. However, due to the high number of informants who turned down the invitations for interviews and limited collection time, convenient sampling was subsequently used, which brings with it limitations associated with ad hoc approaches and a tendency to affect evidence quality [29]. The study context was limited to Lusaka out of 117 districts in Zambia due to COVID-19 pandemic travel restrictions, which has a potential to limit the transferability of the findings.

## Conclusion

Our study highlights private providers' awareness of the NHI and their need to understand its nature and operations. Overall, private providers perceived themselves as valuable to the NHI in complementing public facilities but had different opinions regarding inclusion in the NHI. Their major contributions to the NHI include increased access to drugs in pharmacies, advanced testing and diagnoses in diagnostic centres and specialized care in hospitals and clinics. Besides, private providers raised several conditions necessary for them to engage in the scheme, perceived engaging in the NHI as an opportunity but with challenges and expressed a desire for sustainable NHI management. Considering these aspects can help align modalities regarding consensual agreements between NHIMA and private providers in leveraging the private sector to achieve UHC. Furthermore, private providers emphasized the importance of having more dialogue with the NHI authority to address their concerns during their inclusion in the scheme. This can be considered as the scheme continues to be rolled-out in Zambia. Studies that scale the findings in this study to private providers throughout the country and those that can collect perceptions about the NHI from other stakeholders such as the NHI insured members can further the scope of this research in providing inputs for the implementation of the NHI.

## Supporting information

**S1 Appendix. Interview guide.**
(DOCX)

## Acknowledgments

We acknowledge the private providers who participated in this study and the Clinton Health Access Initiative for funding this research.

## Author Contributions

**Conceptualization:** Kwangaika Mwala Sinjela, Warren Mukelabai Warren Simangolwa, Lindsey Hehman, Mpuma Kamanga, Wesley Kapaya Mwambazi, Jesper Sundewall.

**Data curation:** Kwangaika Mwala Sinjela.

**Formal analysis:** Kwangaika Mwala Sinjela.

**Funding acquisition:** Lindsey Hehman, Jesper Sundewall.

**Investigation:** Kwangaika Mwala Sinjela.

**Methodology:** Kwangaika Mwala Sinjela.

**Project administration:** Lindsey Hehman.

**Supervision:** Jesper Sundewall.

**Validation:** Warren Mukelabai Warren Simangolwa, Lindsey Hehman, Mpuma Kamanga, Wesley Kapaya Mwambazi, Jesper Sundewall.

**Visualization:** Kwangaika Mwala Sinjela, Warren Mukelabai Warren Simangolwa, Jesper Sundewall.

**Writing – original draft:** Kwangaika Mwala Sinjela.

**Writing – review & editing:** Kwangaika Mwala Sinjela, Warren Mukelabai Warren Simangolwa, Lindsey Hehman, Mpuma Kamanga, Wesley Kapaya Mwambazi, Jesper Sundewall.

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
