## [Decision Letter · Decision Letter 0]

12 May 2022

Exploring for-profit healthcare providers’ perceptions of inclusion in the Zambia National Health Insurance Scheme: A qualitative content analysis"

PONE-D-21-13577

Dear Dr. Sundewall,

We’re pleased to inform you that your manuscript has been judged scientifically suitable for publication and will be formally accepted for publication once it meets all outstanding technical requirements.

Kind regards,

Sabeena Jalal, MBBS, MSc, MSc, SM

Academic Editor

PLOS ONE

Additional Editor Comments (optional):

Qualitative research articles are far and few. Thank you for submitting this work. This article explored the for-protfit healthcare providers' perceptions of inclusion in the Zambua National Health Insurance Scheme. A qualitative content analysis. I was glad to see in the references a paper by Francis-Xavier et al, PLoS ONE (2019) about Ghana. Literature search was conducted well. I appreciate the limitation of generalization of the findings has been expressed well in line 313. There is at present a debate about private and public providers. It was useful to see this discussed in the paper. Education of the providers and helping them understand the scheme are practical solutions to the on ground problems. Such as unclear communication on ground. This manuscript is addressing an important issue. 

Some people might say that being a qualitative research, this study is weak. however, the reviewers and I feel that this study is needed and it adds to the literature.

Reviewers' comments:

Reviewer's Responses to Questions

**Comments to the Author**

1. Is the manuscript technically sound, and do the data support the conclusions?

Reviewer #1: Yes

Reviewer #2: Yes

2. Has the statistical analysis been performed appropriately and rigorously? 

Reviewer #1: N/A

Reviewer #2: Yes

3. Have the authors made all data underlying the findings in their manuscript fully available?

Reviewer #1: Yes

Reviewer #2: Yes

4. Is the manuscript presented in an intelligible fashion and written in standard English?

Reviewer #1: Yes

Reviewer #2: Yes

5. Review Comments to the Author

Reviewer #1: This is a relevant topic.

Please correct the title: Exploring for-profit healthcare providers’ perceptions of inclusion in the Zambia

National Health Insurance Scheme: A qualitative content analysis"

Was NVIVO used for analysis or only Microsoft word and excel? If NVIVO was not used, please explain in the manuscript why.

Please make a table for the themes identified and the codes.

Please make a word cloud figure for the manuscript.

Please format the paper according to the PLOS One author guidelines. There are at present too many headings, and the PLOS One format has not been followed.

Please add a section on limitations.

Thanks

Reviewer #2: It is an interesting topic. It is well written. This article is well research and will add new information to the available literature. I was particularly interested in private providers need to understand government initiatives that require their engagement. Providers that were undecided or did not want to participate emphasized the need to understand the scheme. On the other hand, those found to be sufficiently aware and decided whether or not to accredit were those in direct contact with NHIMA.

6. PLOS authors have the option to publish the peer review history of their article (what does this mean?). If published, this will include your full peer review and any attached files.

Reviewer #1: No

Reviewer #2: No

---

## [Editor Report · Acceptance letter]

18 May 2022

PONE-D-21-13577 

Exploring for-profit healthcare providers’ perceptions of inclusion in the Zambia National Health Insurance Scheme: A qualitative content analysis 

Dear Dr. Sundewall:

I'm pleased to inform you that your manuscript has been deemed suitable for publication in PLOS ONE. Congratulations! Your manuscript is now with our production department. 

Kind regards, 

on behalf of

Dr. Sabeena Jalal 

Academic Editor

PLOS ONE